# Do Host Plant and Associated Ant Species Affect Microbial Communities in Myrmecophytes?

**DOI:** 10.3390/insects10110391

**Published:** 2019-11-06

**Authors:** Mario X. Ruiz-González, Céline Leroy, Alain Dejean, Hervé Gryta, Patricia Jargeat, Angelo D. Armijos Carrión, Jérôme Orivel

**Affiliations:** 1Departamento de Ciencias Biológicas, Universidad Técnica Particular de Loja, San Cayetano Alto s/n, Loja 1101608, Ecuador; 2AMAP, IRD, CIRAD, CNRS, INRA, Université de Montpellier, 34000 Montpellier, France; celine.leroy@ird.fr; 3CNRS, UMR EcoFoG, Agroparistech, CIRAD, INRA, Université de Guyane, Université des Antilles, Campus Agronomique, 97379 Kourou, France; 4Ecolab, Université de Toulouse, CNRS, INPT, UPS, 31062 Toulouse, France; alain.dejean@wanadoo.fr; 5Laboratoire Evolution & Diversité Biologique (EDB UMR 5174), CNRS, IRD, Université de Toulouse, 31062 Toulouse, France; herve.gryta@univ-tlse3.fr (H.G.); patricia.jargeat@univ-tlse3.fr (P.J.); 6Biodiversity Genomics Team, Plant Ecophysiology & Evolution Group, Guangxi Key Laboratory of Forest Ecology and Conservation, College of Forestry, Daxuedonglu 100, Nanning 530005, Guangxi, China; adcarrion86@gmail.com

**Keywords:** microbial diversity, domatia, *Allomerus decemarticulatus*, *Allomerus octoarticulatus*, *Azteca* sp. cf. *depilis*, *Cordia nodosa*, *Hirtella physophora*

## Abstract

Ant-associated microorganisms can play crucial and often overlooked roles, and given the diversity of interactions that ants have developed, the study of the associated microbiomes is of interest. We focused here on specialist plant-ant species of the genus *Allomerus* that grow a fungus to build galleries on their host-plant stems. *Allomerus*-inhabited domatia, thus, might be a rich arena for microbes associated with the ants, the plant, and the fungus. We investigated the microbial communities present in domatia colonised by four arboreal ants: *Allomerus decemarticulatus*, *A. octoarticulatus*, *A. octoarticulatus* var. *demerarae*, and the non-fungus growing plant-ant *Azteca* sp. cf. *depilis*, inhabiting *Hirtella physophora* or *Cordia nodosa* in French Guiana. We hypothesized that the microbial community will differ among these species. We isolated microorganisms from five colonies of each species, sequenced the 16S rRNA or Internal TranscribedSpacer (ITS) regions, and described both the alpha and beta diversities. We identified 69 microbial taxa, which belong to five bacterial and two fungal phyla. The most diverse phyla were Proteobacteria and Actinobacteria. The microbial community of *Azteca* cf. *depilis* and *Allomerus* spp. differed in composition and richness. Geographical distance affected microbial communities and richness but plant species did not. Actinobacteria were only associated with *Allomerus* spp.

## 1. Introduction

The emergence of social life promoted the ecological success of the social species, but it also had profound evolutionary, ecological, and economic impacts on many other species, thus shaping life on Earth [1,2,3]. Indeed, the dominance of social species can have important consequences for the functioning of ecosystems and biodiversity in general [4]. On the other hand, sociality is associated with costs linked to the many potential risks associated with living in a group [5,6,7]. Thus, one can expect that both fitness benefits and costs shape the communities of organisms associated with social species.

Social insects, particularly ants, are good examples of these successful social organisms. They are considered as ecological engineers involved in many key ecosystem processes [2,8] and, given the high relatedness within colonies, the spread of pathogens and parasites can have strong deleterious effects both at the individual and colony levels [9]. Therefore, ant-associated microorganisms have received particular attention, mainly focusing on the complex multipartite networks of interactions among fungus-growing ants, their associated fungi, and a diversity of both detrimental and beneficial associated microorganisms [3,10,11,12,13,14,15,16,17,18]. Non fungus-growing ants also exhibit the ability to modify both abiotic and biotic characteristics, thus selecting for different microbial communities while boosting microbial diversity in their nests [19,20,21,22].

The recent discovery of bacteria and fungi associated with plant-ants (i.e., ants associated with myrmecophytes or plants providing them with a nesting place in the form of hollow structures called “domatia”) shed light on an overlooked role of microbiomes in ant-plant interactions [23,24,25,26]. It has also contributed to improving our understanding of how ants regulate their immediate environment and, thus, how they affect the diversity and functioning of the ecosystem. Moreover, myrmecophytes constitute a robust system that can be studied in order to answer such questions, since they are most often inhabited by one or a few specialized plant-ant species, with usually one colony per plant [27,28,29]. In addition, the environmental conditions provided by myrmecophytes can be considered as different from the surroundings, so that the microorganisms found inside the domatia are expected to differ according to the identity of both the associated ant species and the plant. As a consequence, both the local environment provided by the plant and the traits of the ant species might affect the diversity and composition of the associated microbial communities, which can be considered as a selection force or niche-filtering [30,31].

Here we investigated the bacterial and fungal microorganisms associated with two sympatric ant-plants, *Cordia nodosa* Lamarck (Boraginaceae) and *Hirtella physophora* Martius and Zuccharini (Chrysobalanaceae), and their associated ants at two sites in French Guiana. These myrmecophytes are mainly inhabited by ants of the genus *Allomerus* (Myrmicinae), but *C. nodosa* can also host *Azteca* sp. cf. *depilis* (Dolichoderinae) [32]. Ants of the genus *Allomerus* are specific plant-ants [33] that have developed a particular behaviour for prey capture, which relies on the construction of galleries on their host plants to ambush prey [34]. To this end, they have evolved the practice of a novel kind of fungal agriculture with non-nutritional purposes, which involves highly fine-tuned multipartite plant-ant-fungus-bacteria associations [26,35,36,37]. We isolated and identified bacteria and fungi present inside the domatia of the two plants inhabited by the different ant species and hypothesized that a strong niche filtering affected the microbial diversity and composition. That is, that the microbial community might be influenced by the location of the sampling site, and that both the species of the plants and the ants should influence bacteria and fungi present inside the domatia.

## 2. Material and Methods

### 2.1. Sampling

Samples were collected at two locations in French Guiana, Montagne des Singes (MdS: 5°04’21.92” N; 52°41’51.13” O) and Basevie (Bsv: 5°05’21.91” N; 53°01’28.39” O), between January and February 2009. Both sites were located about 40 km from each other. It should be noted the microbial isolation, culture and sequencing were performed just after the samples were collected (see methods below), while microbial identifications have been updated more recently, thus reflecting the potential changes in the GenBank database, which could have occurred since the sampling period. We randomly selected five colonies of each of four plant-ant species that inhabit the domatia of two myrmecophytic plant species: *Cordia nodosa* Lamarck (Boraginaceae) and *Hirtella physophora* Martius et Zuccharini (Crysobalanaceae). In the Montagne des Singes area, *C. nodosa* is colonized by either *Allomerus octoarticulatus* var. *demerarae* or *Azteca* sp. cf. *depilis*, and *H. physophora* is colonized both by *A. decemarticulatus* or *A. octoarticulatus*. Note that the two *A. octoarticulatus* species are different, cryptic species, and each is associated to a single host plant species, *H. physophora* or *C. nodosa*, in French Guiana. These two *A. octoarticulatus* species cannot be separated based on morphology alone. However, they separate into monophyletic sister clades based on the barcoding of the COI (Cytochrome c oxidase subunit I) gene fragment [38]. *Allomerus octoarticulatus* var. *demerarae* appears always and solely associated with *C. nodosa*, whatever its geographic origin, while *A. octoarticulatus* can be associated with a variety of host plants over its distribution range, although only found inhabiting *H. physophora* in French Guiana.

In Basevie, *C. nodosa* was colonized only by *A. octoarticulatus* var. *demerarae* and *H. physophora* only by *A. decemarticulatus*.

The *C. nodosa* domatia are located in the stems below each sympodial fork [39], while in *H. physophora* they result from the curling under of the leaf margin on either side of the petiole [40]. For each of the 30 plants with well-established ant colonies, we collected three leaves with domatia from the upper part of the plants to minimize the potential contamination of the leaves with soil microbiota. Leaves were collected from opposite branches. We sampled leaves of similar age based on their colouration. Then, domatia were individually transported in sterile zip bags to the laboratory. Each domatia was dissected with flame sterilized forceps and scalpel. We used 100 μL of sterile physiological saline solution (0.90% w/v NaCl) to thoroughly wash the inner walls and collect as many microorganisms as possible. Each sample was stored at 4 °C in 1.5 mL Eppendorf vials.

### 2.2. Microbial Isolation and Identification

Dilutions of the samples (1/10^6^) were prepared after being gently vortexed, and 50 μL of dilution were plated on solid MYG medium (1% malt extract, 0.4% yeast extract, 0.4% glucose, 1.5% agar). Two plates were inoculated by sample, and the cultures were kept in dark conditions at 20 °C for up to 15 days. Afterward, we selected random bacterial and fungal colonies belonging to every potentially different morphospecies based on colony colour, size, and shape. Fungi were transferred to new plates to obtain pure cultures.

Fungal DNA was extracted from mycelium pieces with the Chelex^®^ method [41], and we used direct PCR of intact bacteria as template. The 16S rRNA region of bacteria was amplified using the FD1 and RP2 primers [42] and the fungal ITS region of the rRNA with ITS1 and ITS4 primers [43]. PCR products were sequenced by Genoscreen (Lille, France), and edited with Chromas 2.6.5 (Technelysium Pty Ltd, Brisbanes, Australia). We checked for the closest sequences in GenBank [44] by following a BLAST procedure.

### 2.3. Alpha Diversity

Species richness was calculated as the total number of microbial species present in each community (*S*), and species abundance as the total number of microbial isolates of each species that appeared in our samples (*N*). To take into account sample size, we calculated the Margalef’s index of species richness ($D_{MG}=\frac{S-1}{LnN}$) obtained with the ‘vegan’ R package. Then, to describe the alpha diversity, we obtained five indexes with the ‘phyloseq’ R package: Chao (± SE), Shannon, Simpson, Inversed Simpson, and Fisher. We also calculated the percentage of completeness as $C\left(\%\right)=\frac{S}{Chao}$. All calculations were conducted for Site (Basevie and Montagne des Singes), Plant (*C. nodosa* and *H. physophora*), Ant (*A. decemarticulatus*, *A. octoarticulatus*, *A. octoarticulatus* var. *demerarae* and *Azteca* sp. cf. *depilis*), and their interactions of Site × Plant, Site × Ant, Plant × Ant, and Site × Plant × Ant. The values of the Shannon indexes between pairs of communities were compared using Student’s *t*-test, and *p*-values were corrected using a Bonferroni adjustment. Species rarefaction curves were plotted on the expected number of species, and species accumulation curves or sample rarefaction (Mao tau) were computed as a function of the six communities using PAST software [45]. For both curves, the standard errors were converted as 95% confidence intervals.

### 2.4. Beta Diversity

We calculated Bray-Curtis dissimilarities and, then, we clustered together the six microbial communities (Site × Plant × Ant) of domatia with the ‘picante’ R package. Furthermore, to visualize the level of similarity among microbial communities across Site × Plant × Ant and colonies, we conducted multidimensional scaling multivariate data analysis. Then, we conducted a multivariate analysis of variance (MANOVA) using the distance matrix with 999 permutations, and a multilevel pairwise comparison (permutational multivariate analysis of variance, PERMANOVA) to the Bray-Curtis distances with 40,000 permutations, using the ‘vegan’ package from R [46]. Furthermore, to validate the relevance of the pairwise comparisons after the PERMANOVA we performed a pairwise permutation MANOVA with 40,000 permutations, as found in R package ‘RVAideMemoire’.

## 3. Results

### 3.1. Microbial Isolation

A total of 67 bacteria species and two fungi species were isolated, and identified after 16S or ITS sequencing, across all ant domatia (Table 1). Bacteria belonged to the phyla Actinobacteria, Bacteroidetes, Firmicutes, and Proteobacteria. Among the 11 bacterial orders, the best represented were the Rhizobiales (14 species), Enterobacterales (10), Micrococcales (9), Pseudomonadales (9) and Xanthomonadales (9). One Dothideomycetes (Ascomycota) and a Tremellomycetes (Basidiomycota) were also isolated. While Actinobacteria species (9) were not present in *Azteca* sp. cf. *depilis* nests, the two fungal species were only isolated from them. Firmicutes (class Bacilli) species were only present in *A. decemarticulatus*. Three bacteria were present in all ant species across all sites and plants: the insect symbiont *Ochrobactrum* sp. 1, *Sphingomonas echinoides*, and *Luteibacter* cf. *yeojuensis* st 2. Finally, *Arthrobacter* sp. 1 and *Burkholderia* sp. 9 were exclusively associated with *Allomerus* spp. Overall, we found 20 bacterial genera in *A. decemarticulatus* (18 families), *A. octoarticulatus* (14 families), and *Azteca* sp. cf. *depilis* (12 families) domatia, and 19 in *A. octoarticulatus* var. *demerarae* (14 families).

### 3.2. Alpha Diversity

The rarefaction curve and the completeness values suggest that the microbial communities present in *A. octoarticulatus* var. *demerarae* are better characterised compared to the other four communities (Figure 1A,B; Table 2). That is, the slopes of the *C. nodosa* curves appear to be different from the *H. physophora* slopes, and indicate that total microbial diversity would likely be different between the plant species if more samples were collected. We found an overall significantly higher microbial species richness in Montagne des Singes than in Basevie (Table 2 and Table 3). There were no differences in species richness or evenness between plant species, although Chao’s abundance-based richness was higher in *H. physophora* than in *C. nodosa* (Table 2). Microbial richness associated with *Azteca* sp. cf. *depilis* (18) was more than half lower compared with *A. decemarticulatus* (43) or *A. octoarticulatus* (47), and thus exhibited significantly lower species richness and evenness (Table 3). When we analysed the diversity associated with the Site × Plant interaction, we found that *C. nodosa* from Basevie (21 species) had the lowest species richness and diversity, while the highest was found in *H. physophora* (40) from Montagne des Singes (Table 3). When investigating nest microbial diversity associated to Site × Ant communities, we found significant differences in richness in Montagne des Singes between *Azteca* sp. cf. *depilis* and *A. decemarticulatus, A. octoarticulatus,* and *A. octoarticulatus* var. *demerarae* (Table 3). In the interaction Plant × Ant, the only significant differences in species richness were between the microbial communities associated with *Azteca* sp. cf. *depilis* from *C. nodosa*, *A. decemarticulatus* colonising *H. physophora*, and *A. octoarticulatus* var. *demerarae* from *C. nodosa* (Table 3). Finally, in the Site × Plant × Ant interaction, the only microbial communities that exhibited significant differences in richness were those associated with *Azteca* sp. cf. *depilis* in *C. nodosa* and *A. decemarticulatus* in *H. physophora*, both from Montagne des Singes (Table 3).

### 3.3. Beta Diversity

The dendrogram based on Bray-Curtis dissimilarities grouped the two microbial communities of Basevie into a clade, the microbial communities from *Allomerus* domatia in a second clade and the community from *Azteca* sp. cf. *depilis* into a third clade (Figure 2A). Moreover, the clades are related to the phylogenetic diversity of each microbial community (Figure 2B,C). We found overall significant differences among the six microbial communities (*F*_5,89_ = 2.6544; *p*-value = 0.001). The multilevel pairwise comparison detected differences for 10 pairs of communities (Table 3, which were further confirmed by the pairwise permutation MANOVA. The MDS mapping (Figure 3) was well supported by the non-metric fit of *R*^2^ = 0.948, although the linear fit (*R*^2^ = 0.708), and the stress (*S* = 0.228) values were slightly weak.

## 4. Discussion

### 4.1. Microbial Diversity and Composition

The microbial diversity of the studied plant-ant species is similar in genera numbers to the one found associated with fungus-growing ants in their nests [13], even if the taxonomic diversity studied here is based on the microbial ability to grow on media and, thus, the presence of more species can be expected. In another ant-plant associations between *Azteca alfari* and *Cecropia peltata*, Lucas et al. [47] found 22 bacterial phyla across internal and external structures of the plant but 90% of the microbial diversity corresponded to Proteobacteria and Actinobacteria taxa. This study also demonstrated the role that ants play in shaping the composition of microbial communities inside their nests, as it has been shown here too, which can be considered as an effect of niche filtering (see below).

That *Allomerus* ants were associated with the highest number of bacterial species compared to *Azteca* ants, which can be explained either by the fact that they cannot filter as many microbial species as *Azteca* spp., or because they allow and select a wider microbial community composition because of more complex functional roles. It is particularly relevant that only these ants harboured *Actinobacteria* species because the defensive role of some species in fungus gardens of Attine ants is well known [11,48]; and because *Allomerus* ants practise a particular type of fungiculture in their colonies [26]. Recent work investigating the *Azteca* spp. microbiome from myrmecophytic *Cecropia*, found different Actinobacteria species [47] from the ones identified here for *Azteca* sp. cf. *depilis*. One species of Proteobacteria, *Pseudomonas citronellolis*, was recorded as associated with both *Azteca* sp. cf. *depilis* (this study) and *Az. alfari* [47]. This species could be specifically associated with *Azteca* ants. Previous work on bacteria associated with both fungus-growing and arboreal ants detected bacteria from diverse genera that perform different roles, such as defence against parasites and diseases, plant substrate degradation, and nitrogen fixing bacteria from the genera *Burkholderia*, *Curtobacterium*, *Enterobacter*, *Escherichia*, *Pantoea*, and *Rhizobium* [10,13,49,50,51]. We have found bacterial species from all these genera that could perform similar roles. First, they can complement the behaviour of *Allomerus* ants that chew the walls of the domatia to prepare vegetal substrate and culture their mutualistic fungus *Trimmatostroma cordae* [26]. Second, they could act in reinforcing the active transfer of nitrogen to the plant mediated by the mutualistic fungus [36]. Third, they could play a role in the defence of the domatia against diseases by limiting pathogen proliferation.

### 4.2. Variations in Microbial Communities

The observed richness of microbial communities appears driven by both distance (geography) and ant species identity, as a result of niche filtering. That is, the effect of site location in microbial composition means horizontal acquisition of local bacterial species or strains. Furthermore, the effect of ant species in microbial richness suggest the presence of ant-engineered microbial communities. Whether these microbial communities are vertically transmitted or filtered from the environment by the ants remains unclear. Host plant species, however, do not seem to have an effect on richness or evenness of microbial species in domatia. Therefore, in the studied systems, plant species do not contribute to niche filtering. Indeed, the microbial communities associated with *Azteca* sp. cf. *depilis* appeared different from the ones associated with *Allomerus* ants, both in terms of richness and composition. Moreover, the pairwise comparison of the composition of bacterial communities highlighted differences between *A. octoarticulatus* inhabiting *C. nodosa* from Basevie and the other communities but *A. octoarticulatus* from Montagne des Singes. On the one hand, the ant species effect might be related to an active selection of potentially beneficial microorganisms by the ants; or at least the removal of pathogens. It could be also linked to differences in the environmental conditions in the domatia, not related to the plant species, but to the ants. On the other hand, the geographic effect might be driven by dispersal limitations of the microorganisms and/or differences in the local abiotic environment, surrounding plant species or even ant diet. *Allomerus* ants are omnivores; however, because of their particular behaviour in practising a highly specialised type of agriculture, they can be expected to exert a strong selective pressure on the composition of the microbial community such that healthy fungal symbionts are maintained [26]. The lack of bacterial communities specifically associated with host plant species is supported by previous work in acacia-ants [52]. All these results strongly suggest an active role of ants in the assemblage of their associated microbial communities.

## 5. Conclusions

Altogether our results show that the microbial communities present inside the domatia of ant plants are mainly influenced by ant species and, to a lower extent, by the local environment. This suggests that the ants actively select for at least part of the associated microorganisms and, thus, that the latter could have beneficial roles in the survival of the colonies and in the interaction with their host plants. Although the plant does not seem to contribute to the observed diversity in microbial communities, it could indeed benefit from their presence as already shown from the recent studies of Chaetothyriales fungi associated with plant ants. Our understanding of ant-plant-microorganism interactions and of the functioning of these interaction networks appears promising as a mean of shedding more light on the importance of biotic interactions in the evolution of biodiversity.

## Figures and Tables

**Figure 1 insects-10-00391-f001:**
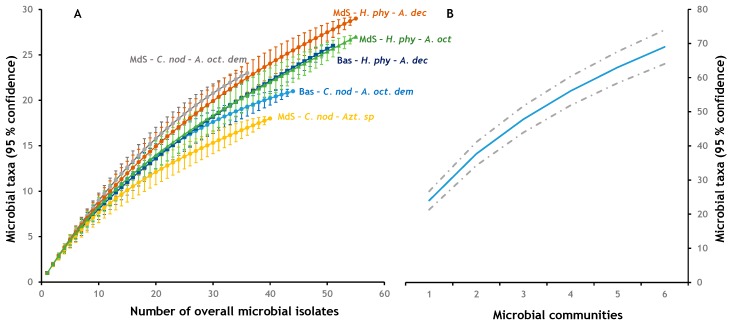
Rarefaction and accumulation curves of microbial species in ant nests. (**A**) Rarefaction curve for each bacterial community; (**B**) species accumulation curve.

**Figure 2 insects-10-00391-f002:**
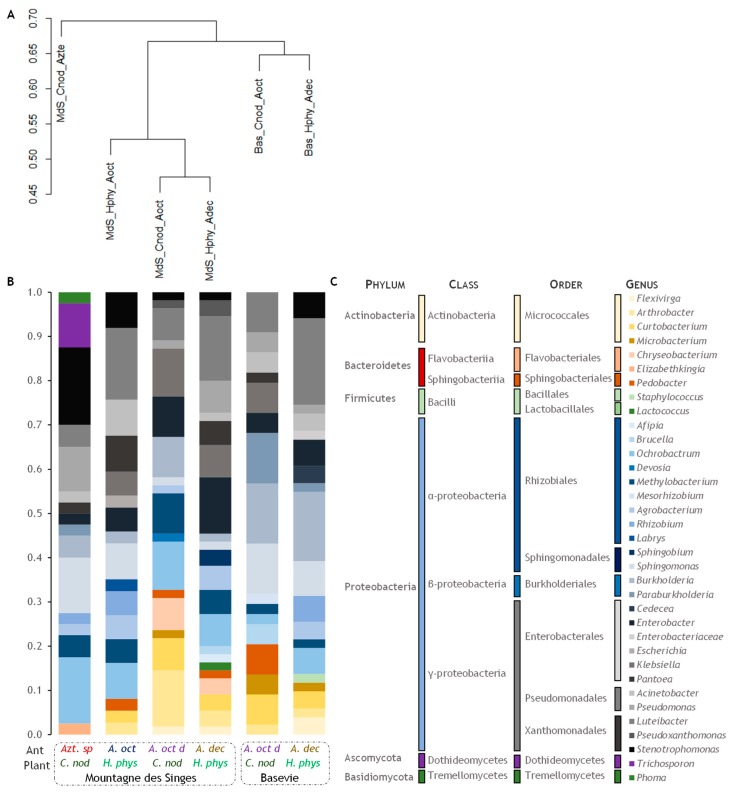
Distribution and abundance of microbial communities in *Cordia nodosa* (C. nod) and *Hirtella physophora* (H. phys) domatia occupied by, *Allomerus decemarticulatus* (A. dec), *A. octoarticulatus* (A. oct), *A. octoarticulatus* var. *demerarae* (A. oct d), and *Azteca* sp. cf. *depilis* (Azt. sp) ant species from Montagne des Singes and Basevie. (**A**) Bray-Curtis dissimilarity dendrogram; (**B**) stackplot of bacterial taxonomic composition; (**C**) microbial diversity by taxonomic level (Phylum, Class, Order, and Genus).

**Figure 3 insects-10-00391-f003:**
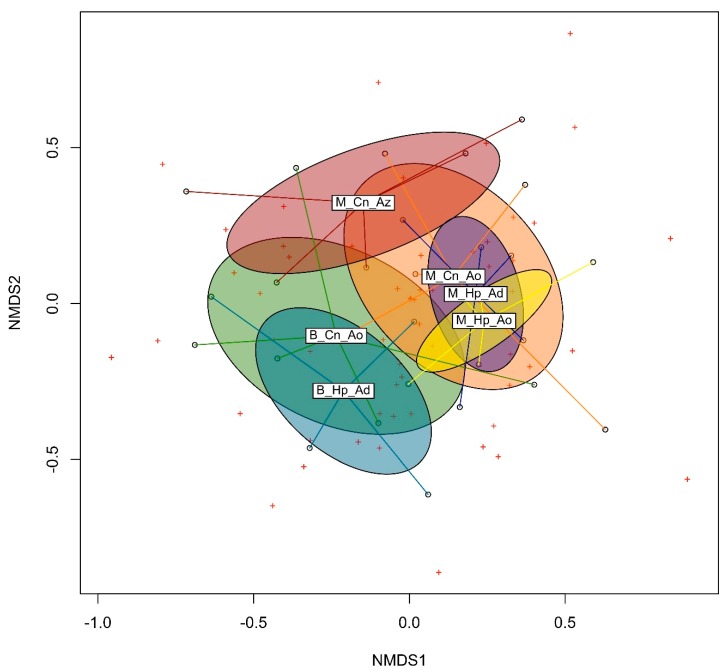
MDS mapping with two dimensions for the six microbial communities. M, Montagne des Singes; B, Basevie; Cn, *Cordia nodosa*; Hp, *Hirtella physophora*; Ad, *Allomerus decemarticulatus*; Ao, *A. octoarticulatus*; Az, *Azteca* sp. cf. *depilis*.

**Table 1 insects-10-00391-t001:** Identification of the taxa isolated from the six communities of arboreal ants and frequency of each species per community. The first column corresponds to the microbial species found in this study with their GenBank accession numbers. Second and third columns provide information about the GenBank taxid closest to our sequences and their percentage of sequence identity. Other columns represent the frequency of each microbial sequence in the studied communities. Azt: *Azteca* sp. cf. *depilis*; Ao: *Allomerus octoarticulatus*; Aod: A. *octoarticulatus* var. *demerarae*; Ad: *A. decemarticulatus*; M: Montagne des Singes; B: Basevie; C: *C. nodosa*; H: *H. physophora*.

Microbial Species in Domatia (New GENBANK Accession) Phylum–Class–Order–Family–Species	Closest Taxid Accession nb	Identity (%)	Azt M_C	Ao M_H	Aod M_C	Ad M_H	Aod B_C	Ad B_H
Actinobacteria–Actinobacteria–Micrococcales–Dermacoccaceae								
*Flexivirga* sp. (MN437546)	MH699193	97.56		0.018		0.019		0.039
Actinobacteria–Actinobacteria–Micrococcales–Microbacteriaceae								
*Arthrobacter* sp. 1 (MN437547)	KX036592	99.00		0.127	0.027	0.037	0.023	0.020
*Curtobacterium albidum* (MN437548)	MK414948	99.71			0.027	0.019		
*Curtobacterium flaccumfaciens* st 4 (MN437549)	KT159381	99.54					0.045	0.020
*Curtobacterium luteum* st 2 (MN437550)	JQ660282	99.70		0.055		0.019		0.020
*Curtobacterium* sp. 1 (MN437551)	MK704290	99.90					0.023	
*Curtobacterium* sp. 9 (MN437552)	MH915626	98.31		0.018				
*Microbacterium* sp. 1 (MN437554)	JX566640	97.63						0.020
*Microbacterium* sp. 3 (MN437553)	MK578285	99.79		0.018			0.045	
Bacteroidetes–Flavobacteriia–Flavobacteriales–Flavobacteriaceae								
*Chryseobacterium* sp. 1 (MN437556)	LT547832	97.00		0.055				
*Chryseobacterium jejuense* (MN437555)	KM114947	98.00		0.018		0.037		
*Elizabethkingia* sp. 2 (MN437557)	KP975262	95.30	0.025					
Bacteroidetes–Sphingobacteria–Sphingobacteriales–Sphingobacteriaceae								
*Pedobacter* sp. 1 (MN437558)	KP708597	98.10				0.019	0.023	
*Pedobacter* sp. 2 (MN437559)	AB461805	96.66		0.018	0.027		0.045	
Firmicutes–Bacilli–Bacillales–Staphylococcaceae								
*Staphylococcus haemolyticus* (MN437560)	MF157599	99.71						0.020
Firmicutes–Bacilli–Lactobacillales–Streptococcaceae								
*Lactococcus lactis* (MN437561)	MF972078	99.46				0.019		
Proteobacteria–Alphaproteobacteria–Rhizobiales–Bradyrhizobiaceae								
*Afipia* sp. 1 (MN437562)	KY827230	99.00				0.019		
Proteobacteria–Alphaproteobacteria–Rhizobiales–Brucellaceae								
*Brucella* sp. (MN437563)	CP007717	92.54				0.019	0.045	
*Ochrobactrum* sp. 1 (MN437564)	AY914071	100.00	0.150	0.109	0.081	0.074	0.023	0.059
Proteobacteria–Alphaproteobacteria–Rhizobiales–Hyphomicrobiaceae								
*Devosia* sp. 2 (MN437565)	LC317339	99.27		0.018				
Proteobacteria–Alphaproteobacteria–Rhizobiales–Methylobacteriaceae								
*Methylobacterium* cf. *persicinum* (MN437568)	NR_041442	99.60	0.050	0.073	0.054	0.037	0.023	
*Methylobacterium* cf. *phyllostachyos* (MN437567)	FR872484	99.80		0.018		0.019		0.020
*Methylobacterium* sp. 5 (MN437566)	KC702828	99.72					0.023	
Proteobacteria–Alphaproteobacteria–Rhizobiales–Rhizobiaceae								
*Agrobacterium* sp. (MN437571)	EU295450	98.64	0.025	0.018	0.054	0.037		0.020
*Agrobacterium tumefaciens* st 2 (MN437570)	KU240580	99.68						0.020
*Agrobacterium tumefaciens* st 3 (MN437569)	KY874047	99.62				0.019		
*Rhizobium* sp. 1 (MN437573)	LC385714	98.29			0.027			
*Rhizobium* sp. 4 (MN437574)	KP219134	99.71			0.027			0.020
*Rhizobium* sp. 5 (MN437572)	MH327921	97.54	0.025					0.039
Proteobacteria–Alphaproteobacteria–Rhizobiales–Xanthobacteraceae								
*Labrys* sp. 1 (MN437575)	KR778886	99.72			0.027			
Proteobacteria–Alphaproteobacteria–Sphingomonadales–Sphingomonadaceae								
*Sphingobium* sp. (MN437576)	HM321152	98.27				0.037		
*Sphingomonas echinoides* (MN437577)	MH725538	98.90	0.125	0.018	0.054	0.019	0.114	0.078
*Sphingomonas polyaromaticivorans* st 1 (MN437578)	HM241216	99.63			0.027			
Proteobacteria–Betaproteobacteria–Burkholderiales–Burkholderiaceae								
*Burkholderia contaminans* (MN437582)	KY886142	99.80	0.050					
*Burkholderia* sp. 5 (MN437583)	AB299574	98.98		0.055			0.068	
*Burkholderia* sp. 9 (MN437580)	KX232126	98.78		0.018	0.027	0.019	0.045	0.020
*Burkholderia* sp. 21 (MN437581)	JN634250	96.51		0.018				
*Burkholderia tropica* st 1 (MN437579)	KT390912	99.67					0.023	0.137
*Paraburkholderia fungorum* (MN437584)	MG576012	99.71	0.025				0.114	0.020
Proteobacteria–Gammaproteobacteria–Enterobacterales–Enterobacteriaceae								
*Cedecea lapagei* (MN437585)	MH074798	99.39						0.039
*Enterobacter hormaechei* cf subsp *xiangfangensis* (MN437588)	MG928407	99.16		0.018			0.045	0.039
*Enterobacter* sp. 1 (MN437587)	JQ660160	99.52	0.025	0.055	0.054	0.130		0.020
*Enterobacter* sp. 5 (MN437586)	KM021337	98.58		0.018				
*Enterobacteriaceae* sp. 2 (MN437589)	KJ934757	98.55						0.020
*Escherichia* sp. (MN437590)	MH465145	99.13			0.027			
*Klebsiella aerogenes* st 1 (MN437591)	JF494822	99.25		0.091	0.054	0.074	0.068	
*Klebsiella oxytoca* (MN437592)	KT260783	98.90		0.018				
Proteobacteria–Gammaproteobacteria–Enterobacterales–Erwiniaceae								
*Pantoea agglomerans* (MN437593)	AF130896	98.90	0.025		0.054	0.056	0.023	
*Pantoea dispersa* st 1 (MN437594)	KC182050	98.11			0.027			
Proteobacteria–Gammaproteobacteria–Pseudomonadales–Moraxellaceae								
*Acinetobacter* cf. *bereziniae* (MN437596)	MK087738	100.00	0.025		0.081			0.020
*Acinetobacter* sp. 2 (MN437597)	JQ433924	99.27					0.045	0.020
*Acinetobacter* sp. 3 (MN437595)	KR189585	99.30				0.019		
Proteobacteria–Gammaproteobacteria–Pseudomonadales–Pseudomonadaceae								
*Pseudomonas* cf. *aeruginosa* (MN437602)	KT943976	99.62				0.019		
*Pseudomonas* cf. *citronellolis* (MN437599)	JQ659858	99.90	0.100					
*Pseudomonas fulva* (MN437603)	KY511074	98.95				0.019		
*Pseudomonas nitroreducens* (MN437601)	MH675504	98.90					0.045	
*Pseudomonas* sp. 2 (MN437600)	KJ184870	99.90				0.037		0.020
*Pseudomonas* sp. 3 (MN437598)	KM187195	98.33		0.018				
Proteobacteria–Gammaproteobacteria–Xanthomonadales–Rhodanobacteraceae								
*Luteibacter* cf*. rhizovinicus* st 1 (MN437606)	KY938100	99.90						0.020
*Luteibacter* cf*. rhizovinicus* st 2 (MN437607)	EU022023	99.72			0.027			
*Luteibacter* sp.1 (MN437604)	FR714940	99.25		0.018	0.054	0.037		
*Luteibacter* cf*. yeojuensis* st 2 (MN437605)	KF668474	99.90	0.050	0.055	0.081	0.093	0.091	0.176
*Luteibacter* cf*. yeojuensis* st 3 *(MN437608)*	JQ798488	98.74						0.020
Proteobacteria–Gammaproteobacteria–Xanthomonadales–Xanthomonadaceae								
*Pseudoxanthomonas* sp. (MN437609)	MH795540	99.72		0.018		0.037		
*Stenotrophomonas maltophilia* st 1 (MN437612)	MK537385	99.61	0.125		0.027			
*Stenotrophomonas panacihumi* (MN437610)	KF668484	99.35	0.025	0.018	0.054	0.019		0.059
*Stenotrophomonas* sp. 3 (MN437611)	JQ684520	99.50	0.025					
Ascomycota–Dothideomycetes–Pleosporales–Didymellaceae								
*Phoma* sp. (MN435151)	KP307011	99.83	0.100					
Basidiomycota–Tremellomycetes–Trichosporonales–Trichosporonaceae								
*Trichosporon siamense* (MN435152)	AB164370	99.27	0.025					

**Table 2 insects-10-00391-t002:** Alpha diversity indexes: Margalef, Chao, Shannon, Simpson, inversed Simpson, and Fisher. S: species richness, overall number of species recorded; N: total number of microbial isolates in the community; C (%): percentage of completeness.

Communities	S	N	C %	Margalef’s D_MG_	Chao ± SE S	Shannon’s H’	Simpson D	Inversed Simpson D_S_	Fisher α
**Site**									
Basevie	37	95	71.0	7.91	52.11 ± 9.73	3.26	0.959	18.61	22.27
Montagne des Singes	57	186	52.9	10.72	107.75 ± 26.46	3.56	0.946	24.40	28.05
**Plant species**									
*C. nodosa*	42	120	77.8	8.56	54.00 ± 7.96	3.43	0.957	23. 61	22.97
*H. physophora*	53	161	35.9	10.23	147.50 ± 55.56	3.53	0.958	23.72	27.56
**Ant species**									
*A. decemarticulatus*	43	106	73.1	9.01	58.83 ± 9.37	3.42	0.953	21.44	26.94
*A. octoarticulatus*	27	55	15.0	6.49	180.00 ± 74.29	3.02	0.94	16.01	20.98
*A. octoarticulatus demerarae*	35	80	85.4	7.76	41.00 ± 4.53	3.36	0.96	23.70	23.73
*Azteca* sp. cf. *depilis*	18	40	61.5	4.61	29.25 ± 9.53	2.64	0.9125	11.43	12.59
**Site × Plant**									
Basevie × *C. nodosa*	21	44	87.1	5.29	24.11 ± 3.10	2.89	0.950	15.61	15.75
Basevie × *H. physophora*	26	51	48.9	6.36	53.20 ± 18.16	2.93	0.946	13.20	21.24
Montagne des Singes × *C. nodosa*	31	76	50.8	6.93	61.00 ± 20.92	3.15	0.924	18.51	19.53
Montagne des Singes × *H. physophora*	40	110	65.2	8.30	61.38 ± 13.14	3.32	0.936	20.58	22.61
**Site × Ant**									
Basevie × *A. decemarticulatus*	26	51	48.9	6.36	53.20 ± 18.16	2.93	0.953	13.20	21.24
Basevie × *A. octoarticulatus demerarae*	21	44	87.1	5.29	24.11 ± 3.10	2.89	0.912	15.61	15.75
Montagne des Singes × *A. decemarticulatus*	29	55	68.5	6.99	42.33 ± 8.84	3.15	0.924	18.56	24.83
Montagne des Singes × *A. octoarticulatus*	27	55	15.0	6.49	180.00 ± 74.29	3.02	0.94	16.01	20.98
Montagne des Singes × *A. octoarticulatus demerarae*	23	36	77.7	6.14	29.6 ± 5.13	3.05	0.95	19.64	27.45
Montagne des Singes × *Azteca* sp. cf. *depilis*	18	40	61.5	4.61	29.25 ± 9.53	2.64	0.944	11.43	12.59
**Plant × Ant**									
*C. nodosa* × *A. octoarticulatus demerarae*	35	80	85.4	7.76	41.00 ± 4.53	3.36	0.953	23.70	23.73
*C. nodosa* × *Azteca* sp. cf. *depilis*	18	40	61.5	4.61	29.25 ± 9.53	2.64	0.938	11.43	12.59
*H. physophora* × *A. decemarticulatus*	43	106	73.1	9.01	58.83 ± 9.37	3.42	0.958	21.44	26.94
*H. physophora* × *A. octoarticulatus*	27	55	15.0	6.49	180.00 ± 74.29	3.02	0.913	16.01	20.98
**Site × Plant × Ant**									
Basevie × *C. nodosa* × *A. octoarticulatus demerarae*	21	44	87.1	5.29	24.11 ± 3.10	2.89	0.938	15.61	15.75
Basevie × *H. physophora* × *A. decemarticulatus*	26	51	48.9	6.36	53.20 ± 18.16	2.93	0.944	13.20	21.24
Montagne des Singes × *C. nodosa* × *A. octoarticulatus demerarae*	23	36	77.7	6.14	29. 60 ± 5.13	3.05	0.948	19. 64	27.45
Montagne des Singes × *C. nodosa* × *Azteca* sp. cf. *depilis*	18	40	61.5	4.61	29.25 ± 9.53	2.64	0.913	11.43	12.59
Montagne des Singes × *H. physophora* × *A. decemarticulatus*	29	55	68.5	6.99	42.33 ± 8.84	3.15	0.936	18.56	24.83
Montagne des Singes × *H. physophora* × *A. octoarticulatus*	27	55	15.0	6.49	180.00 ± 74.29	3.02	0.924	16.01	20.98

**Table 3 insects-10-00391-t003:** *T*-student tests for Shannon’s H’ and Simpsons’ D. Pairwise comparisons for Site, Plant, Ant, Site × Plant, Site × Ant, Plant × Ant, and Site × Plant × Ant. Bold values denote significant pairwise comparisons. Bas.: Basevie; MdS: Montagne des Singes; C. nod.: *Cordia nodosa*; H. phy.: *Hirtella physophora*; A. dec.: *Allomerus decemarticulatus*; A. oct.: *A. octoarticulatus; A. oct. demer*.: *A. octoarticulatus* var. *demerarae*; Azteca sp.: *Azteca* sp. cf. *depilis*. * denote significant *P*-values after Bonferroni correction.

		Shannon’s Richness			Simpson’s Evenness		
Community 1 vs	Community 2	*t*	df	*p*-Value	*t*	df	*p*-Value
Basevie	Montagne des Singes	2.399	203.420	**0.017 ***	−1.225	143.920	0.223
*C. nodosa*	*H. physophora*	−0.829	274.410	0.408	0.025	264.410	0.980
*A. decemarticulatus*	*A. octoarticulatus*	2.646	119.85	**0.009**	−1.085	104.410	0.281
	*A. octoarticulatus demerarae*	0.498	185.1	0.619	0.423	185.52	0.673
	*Azteca* sp. cf. *depilis*	4.854	83.287	**<0.001 ***	−2.101	57.704	**0.040**
*A. octoarticulatus*	*A. octoarticulatus demerarae*	−2.284	108.14	**0.024**	1.464	87.986	0.147
	*Azteca* sp. cf. *depilis*	2.140	89.375	**0.035**	−1.167	74.564	0.247
*A. octoarticulatus demerarae*	*Azteca* sp. cf. *depilis*	4.562	76.011	**<0.001 ***	−2.397	51.779	**0.020**
Basevie–*C. nodosa*	Basevie–*H. physophora*	−0.225	92.749	0.823	−0.528	87.968	0.599
	MdS–*C. nodosa*	−1.783	104.080	0.077	0.655	85.616	0.514
	MdS–*H. physophora*	−3.059	104.050	**0.003 ***	1.086	70.285	0.281
Basevie–*H. physophora*	MdS–*C. nodosa*	−1.313	98.465	0.192	1.074	74.704	0.286
	MdS–*H. physophora*	−2.387	95.101	**0.019**	1.399	65.154	0.167
MdS–*C. nodosa*	MdS–*H. physophora*	−1.286	172.780	0.200	0.491	154.160	0.624
Basevie–*A. dec.*	Basevie–*A. oct. demer.*	0.225	92.749	0.823	0.528	87.968	0.599
	MdS–*A. dec*	−1.222	100.810	0.225	1.018	86.199	0.311
	MdS–*A. oct.*	−0.475	102.62	0.636	0.606	89.906	0.546
	MdS–*A. oct. demer.*	−0.689	86.952	0.493	1.172	79.185	0.245
	MdS–*Azteca* sp.	1.537	90.593	0.128	−0.463	90.246	0.644
Basevie–*A. oct. demer.*	MdS–*A. dec*	−1.635	98.523	0.105	0.601	94.668	0.550
	MdS–*A. oct.*	−0.780	98.949	0.437	0.090	96.931	0.929
	MdS–*A. oct. demer.*	−1.032	78.551	0.305	0.793	79.779	0.430
	MdS–*Azteca* sp.	1.483	79.961	0.142	−1.081	73.544	0.283
MdS–*A. dec*	MdS–*A. oct.*	0.799	109.76	0.426	−0.517	109.5	0.606
	MdS–*A. oct. demer.*	0.596	87.278	0.553	0.189	88.710	0.850
	MdS–*Azteca* sp.	2.946	87.273	**0.004**	−1.601	70.977	0.114
MdS–A. oct.	MdS–*A. oct. demer.*	−0.220	88.762	0.826	0.712	90.237	0.478
	MdS–*Azteca* sp.	2.14	89.375	**0.035**	−1.167	74.564	0.247
MdS–*A. oct. demer*	MdS–*Azteca* sp.	2.401	75.388	**0.019**	−1.767	65.232	0.082
*C. nodosa*–*A. oct. demer*	*C. nodosa*–*Azteca* sp.	4.562	76.011	**<0.001 ***	−2.397	51.779	**0.020**
	*H. physophora*–*A. decemarticulatus*	−0.498	185.100	0.619	−0.423	185.520	0.673
	*H. physophora*–*A. octoarticulatus*	2.284	108.140	**0.024**	−1.464	87.986	0.147
*C. nodosa*–*Azteca* sp.	*H. physophora*–*A. decemarticulatus*	−4.854	83.287	**<0.001 ***	2.101	57.704	**0.040**
	*H. physophora*–*A. octoarticulatus*	−2.140	89.375	**0.035**	1.167	74.564	0.247
*H. physophora*–*A. dec.*	*H. physophora*–*A. octoarticulatus*	2.646	119.850	**0.009**	−1.085	104.410	0.281
Bas.–*C. nod.*–*A. oct. demer*	Basevie–*H. physophora*–*A. dec.*	−0.225	92.749	0.823	−0.528	87.968	0.599
	MdS–*C. nodosa*–*A. oct. demer.*	−1.032	78.551	0.305	0.793	79.779	0.430
	MdS–*C. nodosa*–*Azteca* sp.	1.483	79.961	0.142	−1.081	73.544	0.283
	MdS–*H. physophora*–*A. dec.*	−1.635	98.523	0.105	0.601	94.668	0.550
	MdS–*H. physophora*–*A. oct.*	−0.780	98.949	0.437	0.090	96.931	0.929
Bas.–*H. phy.*–*A. dec.*	MdS–*C. nodosa*–*A. oct. demer.*	−0.688	86.952	0.493	1.172	79.185	0.245
	MdS–*C. nodosa*–*Azteca* sp.	1.537	90.593	0.128	−0.463	90.246	0.644
	MdS–*H. physophora*–*A. dec.*	−1.222	100.810	0.225	1.018	86.199	0.311
	MdS–*H. physophora*–*A. oct.*	−0.475	102.620	0.636	0.606	89.906	0.546
MdS–*C. nod.*–*A. oct. demer*	MdS–*C. nodosa*–*Azteca* sp.	2.401	75.388	**0.019**	−1.767	65.232	0.082
	MdS–*H. physophora*–*A. dec.*	−0.596	87.278	0.553	−0.189	88.710	0.850
	MdS–*H. physophora*–*A. oct.*	0.220	88.762	0.826	−0.712	90.237	0.478
MdS–*C. nod.*–*Azteca* sp.	MdS–*H. physophora*–*A. dec.*	−2.946	87.273	**0.004**	1.601	70.977	0.114
	MdS–*H. physophora*–*A. oct.*	−2.140	89.375	**0.035**	1.167	74.564	0.247
MdS–*H. phy.*–*A. dec.*	MdS–*H. physophora*–*A. oct.*	0.799	109.760	0.426	−0.517	109.500	0.606

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
