# Peer review of "Do Host Plant and Associated Ant Species Affect Microbial Communities in Myrmecophytes?"

_insects, 2019, doi:10.3390/insects10110391_

Round 1

Reviewer 1 Report

This paper describes the microbial communities associated with  various species of ants and their myrmecophytic host plants in French Guiana.  The authors show that the composition of microbial communities differs among ant species but not among the plant species that house them.  The study addresses an interesting topic and improves our understanding of the ecology of ant-plant associations.  Overall, the research is well-executed and the paper is mostly well-written.  Two major concerns include the analytical approach and the quality of the Discussion text.  These and some minor concerns are explained below.

Major Concerns

1) In general, the paper is well-written.  The Introduction provides a logical, solid natural history framework.  However, the conceptual basis for the paper - environmental filtering - is not introduced until very late in this section (Line 70) and is not adequately developed with supporting citations, etc.  The theory surrounding community assembly in general, and environmental filtering in particular, should be provided in the Introduction to strengthen the conceptual basis for this research.  Likewise, predictions related to potential diversity and evenness differences among plant or ant species are not provided, which suggests that this was an unfocused exploratory project.  The Results text in section 3.2 (alpha diversity) simply reports statistical outcomes.  This section would be much more clear, readable, and interesting if the focus is shifted to biological differences rather than statistical differences.  The conclusion that the C. nodosa rarefaction curves are close to a plateau (lines 149-150) is simply incorrect.  It would be more accurate to say that the slopes of the C. nodosa curves appear to be different from the H. physophora slopes, and indicate that total microbial diversity likely would different between the plant species if more samples were collected.  The length of the Discussion is appropriate, but the ideas are not clearly organized and too much of the text simply reiterates results rather than interpreting them.  The Discussion also does not evaluate potential mechanisms or alternative hypotheses for the observed patterns.  These problems could be corrected without increasing the length of the Discussion by revising the current text to improve clarity and conciseness.

2) The use of multiple t-tests and Bonferroni-adjusted alphas is not ideal, especially given the very large number of t-tests used in this study.  The authors are encouraged either to explore more comprehensive analytical approaches or to refine the question so that the analysis is less of an exploratory exercise.  One possibility would be to include one or more additional variables that are relevant to microbial community structure, such as the mass or estimated surface area of plant tissue in each sample.

Minor Concerns

1) The plant samples were collected 10 years ago.  We can assume that the microbial collection and culture procedures were conducted shortly after the plants were collected, and microbial identifications were completed more recently.  However, this information is not provided.  Given that changes in the Genbank database over the past decade could affect the results of this study, it would be helpful to clarify when different aspects of the work were completed.   

2) I suggest changing the title to better reflect the contribution of the study.  In particular, the possibility that plants or ants "select" for specific microbial communities is not demonstrated by this descriptive study - i.e., the authors show patterns but do not provide mechanisms.

3) Tables 1 and 2 should be moved to online Supplementary Information if possible.

Author Response

Modifications are in green in the revised version of the manuscript

This paper describes the microbial communities associated with  various species of ants and their myrmecophytic host plants in French Guiana.  The authors show that the composition of microbial communities differs among ant species but not among the plant species that house them.  The study addresses an interesting topic and improves our understanding of the ecology of ant-plant associations.  Overall, the research is well-executed and the paper is mostly well-written.  Two major concerns include the analytical approach and the quality of the Discussion text.  These and some minor concerns are explained below.

Major Concerns

1) In general, the paper is well-written.  The Introduction provides a logical, solid natural history framework.  

However, the conceptual basis for the paper - environmental filtering - is not introduced until very late in this section (Line 70) and is not adequately developed with supporting citations, etc.  The theory surrounding community assembly in general, and environmental filtering in particular, should be provided in the Introduction to strengthen the conceptual basis for this research.  Likewise, predictions related to potential diversity and evenness differences among plant or ant species are not provided, which suggests that this was an unfocused exploratory project.

Authors answer: Thanks for pointing out this issue. In fact, we were not dealing solely with environmental filtering in this study, since we also expected the ants to have a significant effect on the associated microbial communities.  Thus, we hypothesized that both abiotic and biotic factors shaped the associated microorganisms, which can be considered as niche filtering. Most of the introduction section is developing this idea and we added a sentence to clearly introduce ‘niche filtering’ as a main working hypothesis.

The Results text in section 3.2 (alpha diversity) simply reports statistical outcomes.  This section would be much more clear, readable, and interesting if the focus is shifted to biological differences rather than statistical differences.

Authors answer: We did not understand the reviewer’s concern here since this section is reporting differences in microbial species richness and diversity between the ant and plant associations. Most of the text is focusing on highlighting lower or higher microbial richness and diversity, which are biological differences.

The conclusion that the C. nodosa rarefaction curves are close to a plateau (lines 149-150) is simply incorrect.  It would be more accurate to say that the slopes of the C. nodosa curves appear to be different from the H. physophora slopes, and indicate that total microbial diversity likely would different between the plant species if more samples were collected.

Authors answer: We have calculated the completeness values to support the output of the rarefaction curves. Moreover, we have added to the text “the slopes of the C. nodosa curves appear to be different from the H. physophora slopes, and indicate that total microbial diversity likely would be different between the plant species if more samples were collected.

The length of the Discussion is appropriate, but the ideas are not clearly organized and too much of the text simply reiterates results rather than interpreting them.  The Discussion also does not evaluate potential mechanisms or alternative hypotheses for the observed patterns.  These problems could be corrected without increasing the length of the Discussion by revising the current text to improve clarity and conciseness.

Authors answer: We have thoroughly revised the discussion section which is now organized in two subsections, one dealing with the microbial richness and composition and the other one with the observed variations and the involved factors. We hope this is making this section clearer for the reader and we also included additional remarks on the potential mechanisms involved.

2) The use of multiple t-tests and Bonferroni-adjusted alphas is not ideal, especially given the very large number of t-tests used in this study.  The authors are encouraged either to explore more comprehensive analytical approaches or to refine the question so that the analysis is less of an exploratory exercise.  One possibility would be to include one or more additional variables that are relevant to microbial community structure, such as the mass or estimated surface area of plant tissue in each sample.

Authors answer: we are aware of the problems associated with multiple comparisons. That is why we have corrected after Bonferroni, a highly conservative procedure. However, we found strong difficulties applying other analytical approaches because not all ant species nest in both plant species. We thank the referee bringing into our attention the possibility to include additional variables. The two variables proposed, however, will be equal for all samples because the procedure to sample for microbial species was replicated across all samples. We can think of applying some kind of GLM method but we should remove some factor as, for example, plant species.

Minor Concerns

1) The plant samples were collected 10 years ago.  We can assume that the microbial collection and culture procedures were conducted shortly after the plants were collected, and microbial identifications were completed more recently.  However, this information is not provided.  Given that changes in the Genbank database over the past decade could affect the results of this study, it would be helpful to clarify when different aspects of the work were completed.   

Authors answer: this is true and we have added this information in the revised version of the manuscript.

2) I suggest changing the title to better reflect the contribution of the study.  In particular, the possibility that plants or ants "select" for specific microbial communities is not demonstrated by this descriptive study - i.e., the authors show patterns but do not provide mechanisms.

Authors answer: we have modified the title, which now reads as “Do host plant and associated ant species affect microbial communities in myrmecophytes?”

3) Tables 1 and 2 should be moved to online Supplementary Information if possible.

Authors answer: we believe these tables are providing important information and should be kept together with the main text. Nevertheless, if space is limiting and the editor agree with your comment, we could move these tables to an online supplementary section.

Reviewer 2 Report

The only significant question I put to the authors is the use of a taxonomic name that is currently a junior synonym. While it could well be the case that there are two species of Allomerus octoarticulatus involved in the study, I do not think the authors have made it clear why they associate one of these with Allomerus octoarticulatus demerarae, thus resurrecting the synonymised name. This is important, because it would be helpful for the authors to indicate how they recognise A. octoarticularus demerarae if this is a cryptic species that is difficult or even impossible to distinguish from other species in the complex.

There is also an interesting questions that I would put to the authors:

Why they think the bacterial communities of two different plant genera in Figure 2 assort by geography rather than plant or ant taxon. Are the insect communities on which the ants are feeding different? are there differences in the chemistry of the soil that would favour certain bacteria or fungi? Including a response (even if speculative) would be valuable.

         Check that all citation numbers are correct  ([46] is not)

Figure 2 has a lot of information, and perhaps could be split up. Other minor questions or emendations are included on the MS.

Author Response

Modifications are in red in the revised version of the manuscript

The only significant question I put to the authors is the use of a taxonomic name that is currently a junior synonym. While it could well be the case that there are two species of Allomerus octoarticulatus involved in the study, I do not think the authors have made it clear why they associate one of these with Allomerus octoarticulatus demerarae, thus resurrecting the synonymised name. This is important, because it would be helpful for the authors to indicate how they recognise A. octoarticularus demerarae if this is a cryptic species that is difficult or even impossible to distinguish from other species in the complex.

Authors answer: A more detailed explanation has been added to justify our separation of A. octoarticulatus into two entities (see in lines 98-104 of the revised version of the manuscript). This separation is based on DNA barcoding of the COI gene fragment and not on morphology because the two species do not show morphological variations.

There is also an interesting questions that I would put to the authors:

Why they think the bacterial communities of two different plant genera in Figure 2 assort by geography rather than plant or ant taxon. Are the insect communities on which the ants are feeding different? are there differences in the chemistry of the soil that would favour certain bacteria or fungi? Including a response (even if speculative) would be valuable.

Authors answer: As shown in Figure 2a, the bacterial communities group together according to the sampling site and not according to the plant or the ant species. That is why we consider that geography plays a role. It could be related to a diversity of factors among which local abiotic environment, surrounding plant species or even ant diet. None of them has been quantified and thus we can only speculate on their respective effects. These ideas have been introduce in the discussion section.

Check that all citation numbers are correct  ([46] is not)

Authors answer: We did not find errors in citation numbers and ref #46 appears to be cited correctly. Note that the citation numbers have changed after revising the manuscript.

Figure 2 has a lot of information, and perhaps could be split up.

Authors answer: We revised the figure to include your suggestions. The figure has been split up into three different panels A, B, and C. Then, we have changed all coloring, so now there is a concordance between the bars and the genera (both in panels B and C). Nevertheless, we kept it as a single figure since it is now clearer thanks to the modifications you suggested.

Other minor questions or emendations are included on the MS.

Authors answer: Most of the suggestions have been taken into account. Please find below our justification for the one for which we did not agree:

Introduction – Eusociality: We did not restrict our argumentation to eusocial organisms because the risks associated with living in a group are inherent to any social species, whether eusocial or not.

Reviewer 3 Report

General comments

This manuscript ' Do host plant and associated ant identities select for 2 microbial communities in myrmecophytes?' written well and I enjoy reading. Thank authors for the nice work.

I have only one minor comments.

In Discussion

Page 16, line 233-235, suggest to modify the sentences ' One species of Proteobacteria, Pseudomonas citronellolis, was present both for Azteca sp. cf. depilis and in Lucas et al. [46] (2019) sampling, and could be specifically associated to Azteca ants.'

to

' One species of Proteobacteria, Pseudomonas citronellolis, was present both in the group of  Azteca sp. cf. depilis and in the sampling list of Lucas et al. [46]. This species could be specifically associated to Azteca ants.'

Author Response

Modifications are in blue in the revised version of the manuscript

This manuscript ' Do host plant and associated ant identities select for 2 microbial communities in myrmecophytes?' written well and I enjoy reading. Thank authors for the nice work.

I have only one minor comments.

In Discussion

Page 16, line 233-235, suggest to modify the sentences ' One species of Proteobacteria, Pseudomonas citronellolis, was present both for Azteca sp. cf. depilis and in Lucas et al. [46] (2019) sampling, and could be specifically associated to Azteca ants.'

to

' One species of Proteobacteria, Pseudomonas citronellolis, was present both in the group of  Azteca sp. cf. depilis and in the sampling list of Lucas et al. [46]. This species could be specifically associated to Azteca ants.'

Authors answer: Thanks for your positive comments.  We modified the sentence, not exactly as suggested, but we tried to make it clearer for the readers.